# Understanding Failure and Improving Treatment Using HDAC Inhibitors for Prostate Cancer

**DOI:** 10.3390/biomedicines8020022

**Published:** 2020-01-30

**Authors:** Zohaib Rana, Sarah Diermeier, Muhammad Hanif, Rhonda J. Rosengren

**Affiliations:** 1Department of Pharmacology and Toxicology, School of Biomedical Sciences, University of Otago, 18 Frederick Street, Dunedin 9016, New Zealand; ranzo073@student.otago.ac.nz; 2Department of Biochemistry, School of Biomedical Sciences, University of Otago, 710 Cumberland Street, Dunedin 9016, New Zealand; sarah.diermeier@otago.ac.nz; 3School of Chemical Sciences, University of Auckland, Private Bag 92019, Auckland 1142, New Zealand; m.hanif@auckland.ac.nz

**Keywords:** HDAC inhibitors, histone deacetylases, epigenetics, prostate cancer, androgen receptor-negative, resistance mechanisms

## Abstract

Novel treatment regimens are required for castration-resistant prostate cancers (CRPCs) that become unresponsive to standard treatments, such as docetaxel and enzalutamide. Histone deacetylase (HDAC) inhibitors showed promising results in hematological malignancies, but they failed in solid tumors such as prostate cancer, despite the overexpression of HDACs in CRPC. Four HDAC inhibitors, vorinostat, pracinostat, panobinostat and romidepsin, underwent phase II clinical trials for prostate cancers; however, phase III trials were not recommended due to a majority of patients exhibiting either toxicity or disease progression. In this review, the pharmacodynamic reasons for the failure of HDAC inhibitors were assessed and placed in the context of the advancements in the understanding of CRPCs, HDACs and resistance mechanisms. The review focuses on three themes: evolution of androgen receptor-negative prostate cancers, development of resistance mechanisms and differential effects of HDACs. In conclusion, advancements can be made in this field by characterizing HDACs in prostate tumors more extensively, as this will allow more specific drugs catering to the specific HDAC subtypes to be designed.

## 1. Introduction

There were ~1.3 million new cases of prostate cancer worldwide in 2018 and it is the second most commonly diagnosed cancer in men [1]. Importantly, it ranks second among all cancer fatalities in males [2]. Even though the five-year survival for localized disease is 100%, it drops down to 29.3% if the prostate cancer metastasizes to other organs [3]. Furthermore, from 2005 to 2018 a 31% increase in the incidence of prostate cancer was observed from 974,000 to 1.3 million new cases [1,4]. The majority of prostate cancers are initially androgen-dependent [5]. In advanced prostate cancer, androgen deprivation therapy, either by chemical castration in the form of luteinizing hormone-releasing hormone agonists/antagonists or surgical castration has typically been used. After a median of 18–24 months of treatment, most patients progress to a metastatic castration-resistant prostate cancer (CRPC) [6,7], which has a median survival time of 23–37 months [2]. Therefore, there is a need for novel therapeutic strategies that exploit the molecular signature of prostate cancers. 

One of these strategies involves modulating histone acetylation and deacetylation mediated by histone acetyl transferases (HATs) and histone deacetylases (HDACs), respectively. HATs and HDACs affect both, histone and non-histone proteins [8]. Histone proteins, H2A, H2B, H3 and H4, form the core of the nucleosomes with DNA wrapped around them to form chromatin [9]. 

The histone core proteins have lysine residues on their N-terminal tails [10]. If these lysines are not acetylated, histones remain positively charged and ionically interact with the negatively-charged phosphate backbone of DNA (Figure 1). In this case, chromatin will remain condensed and the transcription machinery will not be able to access the underlying transcription site. Acetylation of lysine residues leads to chromatin decondensation and transcription initiation [11]. Importantly, the balance in the transcription of oncogenes and tumor suppressor genes is lost in cancer [12]. In addition to histones, non-histone proteins can be acetylated. Acetylation of transcription factors, such as p-53, NF- κB, p50 and PC4, increases their sequence-specific binding to the DNA, whereas acetylation of other factors, such as FoxO1, HMGI (Y) and p65, reduces their ability to bind DNA [8]. Regulation of acetylation via histones and non-histone proteins can have a wide repertoire of effects, including altered gene transcription, DNA damage repair, cell division, signal transduction, protein folding, autophagy and metabolism [13,14]. With such a wide range of actions, understanding the mechanisms of protein acetylation and deacetylation is essential to devise new therapies.

## 2. Histone Deacetylases (HDACs)

Eighteen different types of HDACs have been identified in humans [15]. These are grouped into four different classes based on their homology to yeast and co-factor dependencies. Class I, II and IV are zinc-dependent metalloproteins, whereas class III uses NAD^+^ as a co-factor (Figure 2) [2]. Class II HDACs are longer in length and comprise a deacetylase domain in the C-terminal region, except for HDAC6 which possesses two deacetylase domains toward the N-terminal region [16]. On the other hand, Class I HDACs possess an N-terminal deacetylase domain. Class IV embodies the properties of both class I and class II enzymes with N- and C-terminal HDAC activity [17]. Different members of the HDAC family, their localization within the cell and different functions are shown in Table 1.

Overexpression of HDACs is a recurring theme in different malignancies, including gastric, colon, prostate and breast cancers [11]. In the majority of cases, an overexpression of HDACs is linked to a poor clinical outcome [19]. In a study conducted by Weichert and colleagues, 192 patients diagnosed with prostate cancer after radical prostatectomy were recruited. In total, 134 and 142 cases scored high for HDAC1 and HDAC2 expression, respectively, whereas 182 cases exhibited a high category of staining for HDAC3. Higher Gleason scores which indicate the presence of highly differentiated tumors with a poor prognosis positively correlated with HDAC1 and HDAC2 expression between prostate cancer and normal prostate parenchyma (*p* < 0.006 and *p* < 0.047, respectively). However, HDAC3 levels did not correlate with Gleason scores. The proliferative marker Ki67 also correlated positively with HDAC1 (*p* < 0.032), HDAC2 (*p* < 0.002) and HDAC3 expression (*p* < 0.001). HDAC2 was also significantly associated with relapse-free survival. Although statistically significant data were not obtained for HDAC1 and HDAC3, the results demonstrated that the PSA (Prostate-Specific Antigen)-relapse-free survival was decreased in HDAC-positive cells compared to -negative cells [20]. However, other types of HDAC were not tested in this particular study. Results derived from the microarray expression database for cancers, Oncomine, correspond with results from the aforementioned study conducted by Weichert and colleagues (Figure 3) [21]. Similarly, Figure 3 highlights the changes in gene expression levels of HDAC3, 4, 6, 9 and 10 between prostate carcinoma and normal prostate glands. However, HDAC5, 7 and 11 expression was not increased between prostate carcinoma and normal prostate glands as seen on the Oncomine database [21].

In contrast, only one study has failed to show a difference in the expression of either HDAC1 or HDAC2 between cancerous and normal tissues [55]. Expression of other HDACs (3, 4, 5, 6, 7 and 8) were variable, with five out of the nine sample pairs exhibiting less than 50% expression in tumors as compared to normal cells. None of the tumors exhibited a tumor vs normal counterpart ratio higher than 1.25 for these HDACs. Furthermore, no significant correlation was found between HDAC1 and Gleason scores. However, the sample size (*n* = 24) was lower compared to Weichert et al. as well as results from the Oncomine database. In addition, the patients in this study exhibited a tumor stage of T1 or T2 [55], while in the Weichert et al. study, 47.4% (91 patients), 50.5% (97 patients), 2.1% (4 patients) exhibited T2, T3 and T4-grade tumors, respectively [20]. Thus, the difference in results can be attributed to the smaller sample size and less advanced tumor stages.

## 3. HDAC Inhibitors

As HDAC expression has been shown to be associated with poor clinical outcome, HDAC inhibitors have been explored as a potential therapeutic option. The five classes of HDAC inhibitors include hydroxamic acids, cyclic tetrapeptides, short chain carboxylic acids, benzamides and keto-derivatives [56]. These inhibitors have a well accepted pharmacophore, consisting of a zinc-binding group, coordinating with the zinc ion in the active site, a linker that transverses the active site and a cap for interactions with the external surface [57].

Due to the role of HDAC inhibitors in inducing cell cycle arrest, apoptosis, autophagy, heat shock protein-90 (HSP90) inhibition and reactive oxygen species generation, this class of drugs has been trialed in cancers [58]. In 2006, the FDA approved the use of a hydroxamic acid drug, suberanilohydroxamic acid (SAHA, vorinostat), for the treatment of cutaneous T-cell lymphoma [59]. Since then, HDAC inhibitors, including belinostat, panobinostat and romidepsin, have been approved for peripheral T-cell cutaneous lymphoma and multiple myeloma (Figure 4). Hematological malignancies have shown promising responses to HDAC inhibitors [60]. However, HDAC inhibitors have not successfully cleared clinical trials for solid tumors, despite promising collective results in biologic, preclinical and phase I and II studies [58,61]. For example, in phase III trials for advanced hormone receptor-positive breast cancer, chidamide, a HDAC inhibitor, in combination with exemestane, increased the median progression-free survival to 7.4 months in comparison to 3.8 months with placebo and has been recommended for further testing [62]. Other HDAC inhibitors are undergoing trials in solid tumors, including entinostat in phase III trials for locally advanced or metastatic recurrent hormone receptor-positive breast cancer (Table 2) [63]. 

SAHA, romidepsin, panobinostat and pracinostat (Figure 4) are the four HDAC inhibitors that have undergone clinical trials for CRPC and are discussed below. Apart from romidepsin, SAHA, panobinostat and pracinostat are classed as hydroxamic acid derivatives. There is no clear-cut description of structure activity relationship for testing mainly hydroxamic acid derivatives in the clinical trials conducted for CRPC. Qian and colleagues (2016) recommended entinostat, a benzamide, for clinical testing in prostate cancer based on in vitro and in vivo experiments [67]. Details around the effect of class and structure of HDAC inhibitors remain unexplored. No trends can be observed in the IC_50_ values of different classes of HDAC inhibitors. Hydroxamic acid derivatives, including SAHA, panobinostat, belinostat and pracinostat, inhibit different classes with different efficacies; similarly, different compounds from the benzamide class, including chidamide, entinostat and mocetinostat exhibit different affinities for different HDACs (Table 3). Even while considering a pharmacokinetic parameter such as half-life. Chidamide, entinostat and mocetinostat had a half-life of 16–18, 51.58 and 7–11 h, whereas vorinostat, panobinostat, pracinostat and belinostat had a half-life of 0.8–3.9, 12, 5.6–8.9 and 0.9 h [68,69,70].

### 3.1. Vorinostat (SAHA)

Treatment of CWR22 human prostate xenograft tumors with 25, 50 and 100 mg/kg/day of SAHA administered via intraperitoneal injection for 21 days in male BALB/c nude mice resulted in 78%, 97% and 97% reductions in final mean tumor volumes, respectively [74]. However, these results did not translate to clinical efficacy. Specifically, when an oral dose of 400 mg of SAHA was administered daily to 27 patients with CRPC, 11 patients had to be removed from the trial before 6 months due to toxicities, despite dose reductions [61]. On follow up, 9/11 patients recovered from these toxicities, suggesting that they were drug-induced. Toxicities associated with SAHA were unexpected, as previous trials in a plethora of solid and hematological malignancies used similar doses of 400 mg/day, including the trial that led to its approval for the treatment of cutaneous T-cell lymphoma [75]. However, the safety profile of SAHA in 86 patients with advanced cutaneous T-cell lymphoma required the removal of 9.3% of patients due to cytotoxicity and 10.5% required dose reductions [59]. In another study conducted by Siegel and colleagues (2008), SAHA was examined in 476 patients, where 341 received it as a monotherapy, whereas 135 received it in combination with other treatments. Of the patients who received it as a monotherapy, 8.3% were discontinued, whereas in combinations, 18.5% of patients were discontinued [76]. Another interesting observation was that patients who were removed from the study due to toxicity had higher levels of interleukin-6 (IL-6) at all time points compared to patients who were removed due to progression of cancers [77]. Various studies demonstrated a link between IL-6 levels and non-responsiveness to therapy [61,78]. Given that IL-6 levels were not assessed in the study that led to the approval of SAHA, it will be interesting to see how its activity can be boosted when IL-6 levels are low. Nevertheless, stable disease-free progression was observed in 7% (2/27) of patients treated with SAHA in metastatic CRPC, which lasted 84 and 135 days. No PSA declines >50% were observed. In contrast, 13 patients (48%) were removed due to disease progression. Based on these results, further evaluation using SAHA as a single therapy was not recommended [61].

### 3.2. Panobinostat

Panobinostat (LBH589) was FDA-approved in 2015 for treatment of multiple myeloma in patients who had previously received bertezomib and an immunomodulatory agent [79]. Various preclinical studies showed a promising response with panobinostat in prostate cancer [80,81]. For example, Welsbie and colleagues (2009) demonstrated that panobinostat (20 mg/day, 5 days) prevented the growth of tumors in nude female mice implanted with CWR22v1 cells, whereas control tumors were four times larger after 18 days. Furthermore, panobinostat also reduced PSA to levels below baseline [81]. Two clinical trials with panobinostat as a single agent have been conducted. The first trial involved the administration of an oral dose of 20 mg [82]. Disease progression was seen in 7/8 patients, whereas a >50% decline of PSA was not observed. However, toxicities were minimal. Panobinostat was tolerated well in patients with no recorded grade 4 toxicities. Studies in LnCAP cells showed that lower concentrations of panobinostat can trigger a higher endogenous PSA production, whereas higher concentrations lead to a decrease in endogenous PSA responses [81]. With this in mind, an intravenous infusion (20 mg/m^2^) of panobinostat was administered in clinical trials. Of the 35 patients, 25 reported at least one grade 3 adverse event, whereas grade 4 toxicities were seen in 4 patients. Disease progression occurred in 29/35 patients. Based on these findings, panobinostat was not recommended for further testing.

### 3.3. Romidepsin

Romidepsin belongs to the cyclic tetrapeptide class of HDAC inhibitors [83]. In 2009, it was FDA approved for treating cutaneous T-cell lymphoma [84]. In the study conducted by Molife and colleagues (2010), two studies were cited for the rationale to examine drugs in patients with CRPC [83]. A study conducted by Weichert and colleagues (2008) detailing the overexpression of Class I, II and III HDACs in metastatic CRPC was one [19]. The other study involved testing dacinostat (LAQ824) in patients with colon, breast, skin, prostate, pancreas, liver, sarcoma, thyroid, renal, lung, adrenal and esophageal cancers. The objective of this study was to determine the toxicity of dacinostat using 7 doses (6, 12, 24, 36, 48, 72 and 100 mg/m^2^). The study concluded that dacinostat was well-tolerated [85]. In addition, HSP90 inhibition was shown to occur in response in these patients [85]. HSP90 is a pertinent protein because it has a role in stabilizing the androgen receptor (AR). However, if HSP90 is inhibited by HDAC inhibitors, then it fails to interact with the AR, which prevents AR from getting into a high ligand affinity conformation [83,86].

Efficacy studies using romidepsin in preclinical prostate cancer models have not been conducted. However, a phase II trial has been conducted with romidepsin [83]. The study included 35 patients who were intravenously dosed with 13 mg/m^2^ of romidepsin. Two patients achieved a radiological partial response (RECIST) and greater than 50% decline in PCA lasting more than 6 months. However, 22 patients in the study showed progression, and 11 patients showed stable disease. No grade 4 toxicities were reported; however, 11 patients had to be taken off the treatment regimen early. Based on these results, treatment using romidepsin was not recommended for CRPC.

### 3.4. Pracinostat

Pracinostat, another hydroxamic acid derivative, showed promising activity in in vitro and in vivo cancer models, including colorectal, ovarian and prostate [72,87]. It was also administered orally at a dose of 60 mg 3 times per week for three weeks, in 32 patients with metastatic adenocarcinoma of the prostate [88]. The drug was well tolerated, with most of the patients exhibiting grade 1–2 adverse events. The most common of these were fatigue (34%) and nausea (31%), whereas 15.6% experienced one or more grade 3 events. However, PSA responses were noted only in two patients, which lasted 3 and 21.6 months. The circulating tumor cell (CTC) profile (no. of CTCs/7.5 mL) improved in 9 out of 14 patients [88]. In conclusion, the drug was not recommended solely because of the lack of PSA response. An attempt was also made to assess the ETS-related gene (ERG) and phosphatase and tensin (PTEN) status of patients; however, due to lack of PSA responses, a particular effect of the ERG and PTEN status could not be determined [88].

## 4. Pharmacodynamic Rationale for Treatment Failure Using HDAC Inhibitors

As a single agent, HDAC inhibitors have shown poor activity in CRPC and other solid tumors [2,89]. Previously, this has been attributed to their pharmacokinetic profile [90]. This argument can be made in the case of SAHA, which is classified as a class IV drug in the Biopharmaceutical Classification System, due to its low water solubility (190 µg/mL) and low cell permeability (2 × 10^6^ cm/s) [70]. Among other parameters, the drug exhibits poor oral bioavailability (around 11% and 2% in the dog and rat, respectively) and a short half-life of 12 min. Similarly, panobinostat exhibits poor oral bioavailability of 6% in rats and a moderate bioavailability of 33–50% in dogs, whereas romidepsin has not been administered orally in animals due to poor solubility [91]. However, pracinostat has been classified as a BioPharmaceutical Classification System class I compound, as it has a higher water solubility, higher oral bioavailability (34% and 65% in mice and dogs, respectively) and longer half-life of 4.1 h in dogs [70,92,93]. It is worth noting, however, that while some patients exhibited disease progression, others have shown disease stabilization and PSA declines >50% [61,83,88,94]. Given the conflicting results, the pharmacodynamic profile of HDAC inhibitors is clearly favorable in some patients. 

Recently, new insights have been shed on prostate cancer, bringing the centrality of AR, effects on different classes of HDACs by non-specific HDAC inhibitors and evolution of resistance mechanisms in response to HDAC inhibitors into question.

### 4.1. AR-Negative Prostate Cancers

AR is thought to play a central role in the progression of prostate cancers [95]. Most studies show HDAC-mediated inhibition of HSP-90 acetylation as one of the main mechanisms by which AR can be regulated [83,94]. However, studies report different percentages of AR expression in CRPC. In a study conducted by Leav and colleagues (2001), AR immunostaining was present in more than 95% of metastatic cells irrespective of tumor grade [96]. Pronounced AR staining in the nucleus was also seen in normal prostate secretory and stromal cells [96]. In contrast, a study conducted by Shah and colleagues (2004) indicated that AR expression was variable among patients who died from metastasized CRPC. In this study, AR staining was less than 10% in 41.6% of patients, which suggests that there are alternative AR bypass mechanisms [97]. However, no comparisons were made between different regions of the organs to show differences between adjacent tumor sites [97]. 

These findings are also supported by another study which showed that serum PSA declines of >50% were observed in 56% of 140 patients with metastatic CRPC following treatment with the AR antagonist enzalutamide [98]. The baseline CTC population decreased by 75% of chemotherapy-naïve and 37% of post-chemotherapy patients [97]. Interestingly, tumors classified from the pre-enzalutamide era (1997–2011) differed in molecular signatures from tumor tissues obtained after the approval of enzalutamide [95]. Data pooled from immunohistochemistry and RNA sequencing of 56 patients showed that 88.4% of patients expressed an AR+ prostate cancer, whereas 6.3% had a neuroendocrine (NE) variant and 6.3% had an AR-/NE- variant from the period of 1998-Compared with data from 2012 to 2016, the figure dropped to 63.3% of patients expressing AR, whereas the AR-/NE+ variant rose to 13.3% and the AR-/NE- variant rose to 23.3% [95]. Although enzalutamide-resistant cell lines have not been studied, knock-down models, including the knockdown of the AR in LnCAP cells, were investigated with the aim of understanding the AR bypass mechanisms. The FGF8/MAPK pathway was implicated as a major AR bypass mechanism in cell lines and a patient-derived xenograft model, however, studies in patients pertaining to this pathway have not been conducted [95].

Interestingly, there has been no correlation between AR and PSA, which may just be a trend seen in late stage prostate cancer [95]. However, it has been widely reported that AR is responsible for regulating PSA expression via indirect and direct means [95,99,100,101]. Given the shifting paradigm in the understanding of prostate cancer, the percentage expression of AR requires further investigation. To date, clinical trials have not studied AR expression. It will also be interesting to understand the correlation between AR expression and HDAC inhibition, given that HDAC plays a role in HSP90 acetylation.

### 4.2. Other Cellular Targets of HDACs

Although HDAC inhibitors induce apoptosis and cell cycle arrest in various tumors, they also lead to upregulation of epithelial to mesenchymal (EMT) transitioning proteins and confer protective effects in cancer cells via p21 inductions [22,102,103]. These findings can be attributed to the different functions of HDAC classes, as presented in the studies below. Noteworthily, the HDAC inhibitors, SAHA, romidepsin, panobinostat and pracinostat that underwent trials are not specific and inhibit a range of HDACs (Table 4) [57]. Therefore, rather than inhibiting tumor growth, HDAC inhibitors may facilitate its growth.

Various studies support these theories. Kong and colleagues (2012) showed that HDAC inhibitors caused acetylation of histone 3 residues in the promoter regions of EMT-related factors such as vimentin, zinc finger E-box-binding homeobox 1 (ZEB1), Slug and matrix metalloproteinase (MMP2). The Western blotting analysis also revealed that the levels of HDAC1 and HDAC2 were unchanged and interacted with the promoters of vimentin, Slug and ZEB1 following treatment with trichostatin A (TSA) and SAHA. These findings suggested that the activity of HDACs was repressed, but the amount of HDACs were unchanged [22]. Using immunofluorescence and reverse transcription polymerase chain reaction, an accumulation of vimentin, ZEB1 and F-actin was observed in response to treatments with SAHA and TSA at concentrations of 5 µM and 400 nM, respectively, in PC-3 cells [22]. Following treatment with SAHA or TSA, PC3, Du145 and ARCaPE cells showed an irregular fibroblastoid morphology, reminiscent of EMT. As PC-3 cells are AR- cells, AR+ LnCAP cells were also treated with SAHA (2.5 and 5 µM) and TSA (200 and 400 nM). Increases in the mRNA expressions of ZEB1, Slug, vimentin and N-cadherin were seen, along with increasing protein expression of vimentin and fibronectin 24 h after treatment [22]. Nasopharyngeal, colon, liver and lung cancers have shown similar responses to sodium butyrate, valproic acid (VPA) and SAHA at similar concentrations via upregulations in Snail and Slug [104]. Therefore, active HDAC1 and 2 are capable of inhibiting EMT, and HDAC inhibitors can induce EMT. 

Of the few studies examining changes in angiogenesis elicited by HDAC inhibitors, most of the studies have focused on modulating the expression levels of hypoxia inducible factor-1α (HIF-1α), vascular endothelial growth factor receptor (VEGFR)-2 and vascular endothelial growth factor (VEGF)-A in cancer cell lines [105,106]. Mixed results have been reported in response to different drugs. Specifically, SAHA (2.5–10 µM) reduced VEGFA protein expression in lung cancer cell lines (PC9, HCC827, H1975 and NCI-H460) [105,106]. Interestingly, SAHA (5 mmol/L) and VPA (1 mmol/L) caused sprouting of endothelial cell spheroids from HUVECs via activation of β-catenin [107]. VPA was used at this concentration as it equates to patient plasma levels [107]. A further rationale for these studies was the premise that HDAC5 and HDAC7 induce endothelial cell migration [107]. In another study, SAHA (5 µM) and TSA (400 nM) were able to stop the invasion of HUVECs into type I collagen gel [108]. TSA also inhibited the sprouting of capillaries from ex vivo rat aortic rings and VEGF-induced expression of VEGFR1, VEGFR2 and neuropilin-1 [108]. Furthermore, the VEGF competitive inhibitor, semaphorin, was upregulated by SAHA and TSA in HUVECs [108]. Interestingly, the differential effects of HDAC inhibitors have been linked to their concentrations. Aurora and colleagues (2010) showed that HDAC inhibitors at low concentrations synergized with the competitive inhibitor of angiogenesis, pigment epithelium-derived factor (PEDF), whereas at higher concentrations HDAC inhibitors antagonized PEDF [109]. 

Various class I and II HDACs exert anti-angiogenic effects [16]. Kruppel-like factor 4 (KLF4) expression in breast cancer modulates VEGF activity via HDAC2 and HDAC3. Ray and colleagues (2013) showed that an ectopic expression of individual KLF4, HDAC2 and HDAC3 decreased VEGF-CAT reporter activity in MCF-10A cells [26]. Chromatin immunoprecipitation (ChIP) analysis also revealed that KLF4, HDAC2 and HDAC3 interacted with the VEGF promoter [26]. In another setting, HDAC5 interacted with the promoter of FGF2 and SLIT2, factors that are critical for endothelial cell growth, as determined by microarray analysis and ChIP assays [34]. Accordingly, knockdown of HDAC5 by siRNA led to HUVEC sprouting [109]. Furthermore, silencing of HDAC7, a class II HDAC, led to an up-regulation of the pro-angiogenic factor, platelet derived growth factor-B (PDGF-B) in HUVEC cells [110]. 

The studies discussed above demonstrate that HDACs can in fact exert anti-tumorigenic effects, as seen in studies discussed above. Table 1 sums up the different functions of HDAC subtypes to further reinforce that HDACs can exert both anti-tumorigenic and pro-tumorigenic effects.

### 4.3. Resistance Mechanisms

Resistance to HDAC inhibitors has been explored in preclinical models of various cancers, however, it has not been explored in the clinical setting [111]. Four common themes, P-glycoproteins (P-gp), HDAC upregulation, HAT downregulation and p21 induction, will be analyzed in this review. While these mechanisms have not yet been investigated in this context, understanding their impact has potential to improve prostate cancer treatment using HDAC inhibitors in the future.

### 4.4. P-Glycoprotein

P-gp, encoded by the gene ABCB1/MDR1, is an ATP-dependent efflux pump on the cell membrane which has a broad substrate specificity and expels drugs out of cells [112]. In 1989, Mickley and colleagues showed that the HDAC inhibitor, sodium butyrate, increased the expression of P-gp in four human colon carcinoma cell lines; SW-620, HCT-15, DLD1 and L-180 [113]. These studies have since been repeated using different HDAC inhibitors. SAHA and TSA upregulated P-gp expression in HCT116 colorectal cancer cells within 25 h of exposure [114]. P-gp expression was upregulated at two different steps, translational initiation and mRNA stabilization [115], and HDAC inhibitors increase P-gp expression in both stages [115]. Similarly, Duan and colleagues, 2017 demonstrated an induction in P-gp expression following treatment with HDAC inhibitors, SAHA and TSA, in human placental trophoblast cell lines Bewo and JAR. A significant negative linear relationship between HDAC1/HDAC2 and ABCB1/MDR1 mRNA expression was found after treatment of JAR cells with SAHA and TSA, for 24 h [116]. Transfer of P-gp is another mechanism by which tumor cells can potentially efflux HDAC inhibitors [117]. As P-gp plays a role in preventing the entry of drugs into the brain, the effects of HDAC inhibitors VPA and TSA were tested on a human brain endothelial cell line (hCMEC/D3). Although the expression of VPA and TSA did not induce P-gp expression, it caused mobilization of P-gp between endothelial cells [117]. This non-genetic transfer of P-gp is a novel mechanism by which the functionality of the blood brain barrier is altered [117]. The transfer of P-gp has not been studied in tumor cells, and it will be interesting to see if similar mechanisms are induced in this context. 

The upregulation of P-gp is an impediment to using HDAC inhibitors in combination with other chemotherapeutic agents. Not only are the chemotherapeutic agents such as doxorubicin expelled, but certain HDAC inhibitors are also P-gp substrates [118]. In fact, the EC_50_ values obtained from cytotoxicity assays for romidepsin were proportional to the ABCB1/MDR1 mRNA level (r^2^ > 0.9) [119]. Xiao and colleagues (2005) reported that the efflux of romidespsin is mediated by P-gp. This was proven using a P-gp inhibitor, CsA, which at a concentration of 5 µM reversed the resistance acquired in IGROV1 (human ovarian cancer), MCF7 (breast cancer) and K562 (leukemia) cells to romidepsin. However, in clinical trials with romidepsin, an increased expression of ABCB1/MDR1 transporters was seen in peripheral blood mononuclear cells and circulating tumor cells, but not in tumor biopsy samples [111]. However, SAHA was shown to not be a P-gp substrate, as the percentage cell survival was unchanged in P-gp-positive and P-gp-negative CEM (T-cell leukemia), LoVo (colon carcinoma) and K562 (leukemia) cells [120]. While correlation studies between the ABCB1/MDR1 gene and SAHA were conducted, the expression level of P-gp in the P-gp-positive and -negative cells were not assessed [120]. Additionally, inhibitors of P-gps were not used in combination with SAHA to determine its effect in an environment devoid of P-gp [120]. As evident from the data, SAHA had a differential effect on P-gp-positive and -negative cells at different concentrations. Defining this relationship is likely to be critical, as SAHA has upregulated P-gp in a plethora of studies. Therefore, a newer approach in understanding the effect of HDAC inhibitors on P-gp is needed. This mechanism has not been studied in prostate cancer in relation to treatment with HDAC inhibitors, however, P-gp is active in prostate cancer cells [121].

### 4.5. HDAC Upregulation

Changes in HDAC levels are another proposed mechanism for resistance. In a study conducted by Fiskus and colleagues (2008), HDACs 1, 2 and 4 were upregulated, while HDAC6 was downregulated in an HDAC inhibitor-resistant human leukemia cell line, HL-60/LR [122]. This again is indicative of the different forms of HDACs being overexpressed or activated in the cancer being studied. These resistant cells were established by exposing HL-60 cells to increasing concentrations of LAQ824 (10–200 nM). Interestingly, this cell line was also resistant to SAHA and LBH. Specifically, 10-, 30- and 500-fold increases were seen in IC_50_ values between the non-resistant, HL-60, and the resistant, HL-60/LR, cell line for LAQ824, SAHA and panobinostat, respectively. The doubling time of HL-60 cells was 24 h, whereas HL-60/LR cells had a doubling time of 12 h [122]. This was supported by a significantly higher proportion (62.4%) of HL-60/LR cells in the S phase as compared to 40% of HL-60 cells. The significance of HDAC upregulation was also seen in vivo. Specifically, NOD/SCID mice had a 250% longer survival time after being injected with HL-60 cells as compared to HL-60/LR cells [122]. 

### 4.6. Histone Acetyltransferases (HATs) Downregulation

Acetylation of histone and non-histone proteins is regulated by a balance between HATs and HDACs [2]. A study conducted by Halsall and colleagues (2015) showed that a downregulation of 12 HAT complexes was seen upon treatment of human lymphoid blastoid cells with SAHA and VPA within two hours of treatment [123]. Targets of HAT and HDAC complexes include histone tails at the promoters of active genes; therefore, downregulation of HATs balances the effects of HDAC inhibitors in those regions [123]. However, ChIP-seq analysis revealed that H4K16ac, H3K9ac and H3k27ac did not change dramatically at the transcriptional start site in response to VPA treatment [123]. This can be explained by the downregulation of genes encoding 12 HAT complexes (*MYST2*, *PHF17*, *PHF15*, *ING5*, *BRPF3*, *BRPF3*, *MUST4*, *ING3*, *EPC1*, *YEATS4*, *CSRP2BP*, *TAF6L* and *CREBBP*) in response to SAHA and VPA [123]. Surprisingly, changes in histone methylation were seen as well. Methylation of DNA sequences in the promoter region of genes can lead to transcriptional repression of those genes. A form of methylation, H3K27me3, increased rapidly after 60 min. Therefore, it is thought that this response reverts protein hyperacetylation, counteracting the effects of HDAC inhibitors [89]. Similar results were obtained using VPA in the pluripotent, embryo-derived P19 mouse cell line, where protein and mRNA levels of two HATs (Cbp and p300), were downregulated [124]. Functional assays revealed that total and nuclear activity of Cbp and p300 were also reduced [124]. Surprisingly, acetylation levels, as determined by Western blotting of acetyl-H3, increased following VPA treatment [124]. No explanation was offered in this study, but others have mentioned that a chromatin context may be needed for up- or down-regulation of genes [123]. Histone acetylation, in the study conducted by Halsall and colleagues (2015) was shown to progressively increase in response to VPA and SAHA [123]. Furthermore, similar studies in different environments have not been conducted. Noteworthily, the lymphoblastoid cells used in the Halsall and colleagues (2015) study have high basal acetylation levels [123]. High basal levels of acetylation are undetected in PC3 prostate cancer cells, which suggests that the lymphoblastoid cell model may not be relevant to prostate cancer. However, this mechanism remains unexplored. 

### 4.7. P21 Upregulation

P21 upregulation leads to a decrease in cyclin D1, which is essential for middle G1 phase transitioning [125]. Given that p21 is associated with growth arrest, it should cause tumor growth inhibition. However, in preclinical studies of prostate tumorigenesis, an unexpected outcome was seen. Effects of p21 deletion were observed in a transgenic adenocarcinoma of the mouse prostate group (TRAMP) to mirror prostate tumorigenesis. Specifically, mice with a *p21/Cdkn1a* deletion were crossed with TRAMP mice. Two groups, p21^−/−^ TRAMP and p21^+/−^/TRAMP, were compared to TRAMP mice. Dorsolateral prostates were histopathologically analyzed to assess the different stages of prostate tumorigenesis. In the p21^−/−^ group, no incidence of moderately different (MD) tumors were seen, whereas 1/13 p21^+/−^/TRAMP mice showed MD tumors. In comparison, 5/15, 6/15 and 3/15 TRAMP mice showed well differentiated, MD and poorly differentiated prostate adenocarcinoma lesions [126]. P21 is also associated with poor survival outcomes, higher recurrence and elevated Gleason scores [126]. This contradictory outcome was attributed to the paracrine growth enhancing effect of p21 [126]. A pro-mitogenic effect of p21 in conditioned media was seen in C8 cell line, a mouse embryo fibroblast cell line transformed by E1A and H-Ras [127].

Following HDAC inhibition, p21 levels have been upregulated in various cancers [126,128]. A plethora of studies highlight the role of p21 in cell cycle arrest in response to HDAC inhibition [102,103]. However, the protective role of p21 can be inferred from effects on cells that lack p21. Administration of antisense constructs of p21 in human monocytic U937 cells led to lowered sensitivity to SAHA treatment [129]. Similarly, p21 deficient HCT116 cells were more sensitive to romidepsin as compared to wildtype cells. However, this mechanism remains unstudied in prostate cancer [130].

## 5. Conclusions

Currently, two layers of information are lacking in studies conducted on HDAC inhibitors in cancer. Firstly, the expression profiles of different HDACs in prostate cancer models are absent. Although attempts to study expression profiles have been made, the data are contradictory and incomplete. This is important because different HDACs exhibit opposing pro- and anti-tumorigenic functions. The effect produced by the different HDACs may also lead to the upregulation of different resistance mechanisms. Recently, studies have also shown that HDACs deacetylate non-histone proteins, which in turn can lead to more varied effects [8,131]. Secondly, even though previous studies claim that HDAC inhibitors are capable of inhibiting specific classes of HDACs, information about IC_50_ values with regards to specific HDAC class inhibition is lacking in most reports. In studies where specific HDAC class inhibitions have been measured, IC_50_ values across studies can vary as much as 100-fold [132,133]. The non-selectivity of the drugs can be attributed to the zinc-binding capability of HDAC inhibitors. The reason for this non-selectivity could be that zinc has been estimated to be found in 3000 human metalloproteins which constitutes for 10 percent of the proteome [134]. Examples of zinc containing proteins in their active site include carbonic anhydrase, zinc fingers alcohol dehydrogenase and farnesyltransferase matrix metalloproteinases, protein prenyl trasnferases and metallo-β-lactamases [135]. An alternative approach targeting amino acids at the entry site or the active site tunnel for hydrophobic and π–π interactions could be examined using docking studies and subtype specific HDAC inhibition assays. Therefore, rather than alleviating symptoms, HDAC inhibitors could potentially lead to disease progression.

Another piece of the puzzle that is missing in terms of understanding the role of HDACs in prostate cancer is the involvement of the AR. Recently, the emergence of an AR- prostate cancer has been reported [95,97]. Regulation of AR expression by HDACs have been cited as the main reason for drug trials with HDAC inhibitors. Only a few studies have tried to elucidate pathways in AR-prostate cancers, including the study by Bluemn and colleagues (2017). The question remains as to whether the involvement of certain pathways in AR- prostate cancers make them more or less susceptible to HDAC inhibitors. Since many HDAC inhibitors have failed in the clinic, defining these relationships is important because additional insights would allow for more effective and specific HDAC inhibitors to be rationally designed. These new drugs could potentially be used as either single or combination therapies.

## Figures and Tables

**Figure 1 biomedicines-08-00022-f001:**
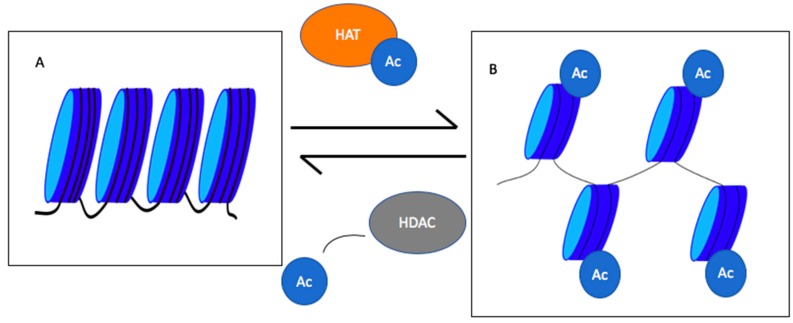
Roles of histone acetyl transferases (HATs) and histone deacetylase (HDACs) in chromatin condensation and decondensation. HATs add an acetyl group to histones decondensing chromatin, whereas HDACs remove acetyl groups leading to chromatin condensation. (**A**) represents the condensed chromatin, and (**B**) shows the decondensed chromatin. Ac = acetyl group [11].

**Figure 2 biomedicines-08-00022-f002:**
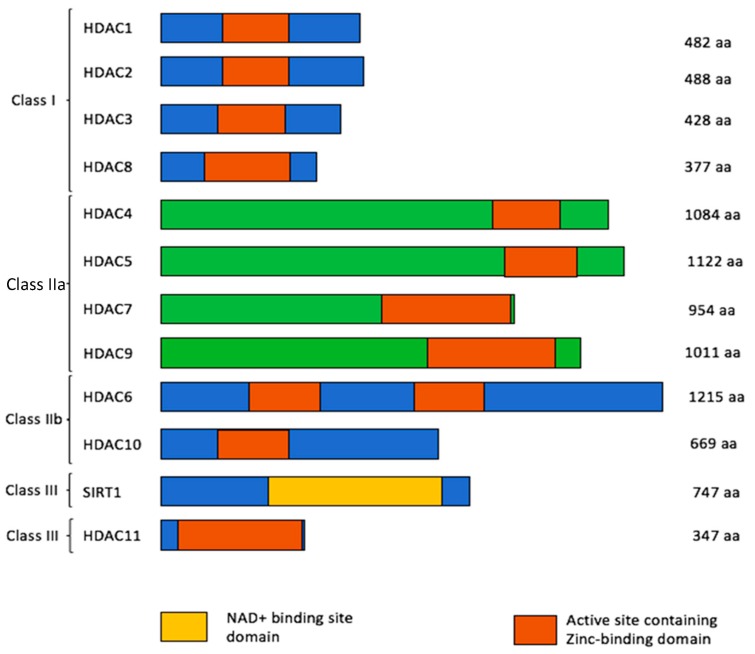
The different classes of HDACs. The color scheme represents the active site containing the zinc-binding domain and NAD+ binding site domain [18].

**Figure 3 biomedicines-08-00022-f003:**
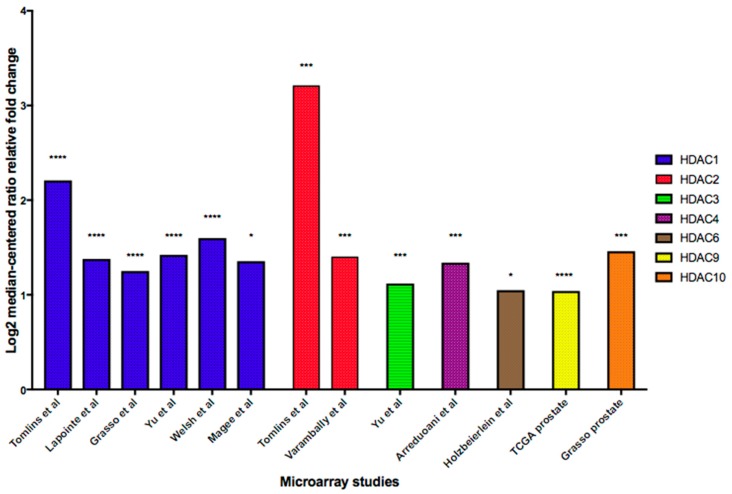
Summary graph of microarray analysis of HDACs from the Oncomine database. The relative-fold change represents the difference in the respective HDAC between prostate carcinoma and normal prostate gland. * *p*-value < 0.05, *** *p* < 0.001 and **** *p* < 0.0001. This figure shows a summary of Appendix A.

**Figure 4 biomedicines-08-00022-f004:**
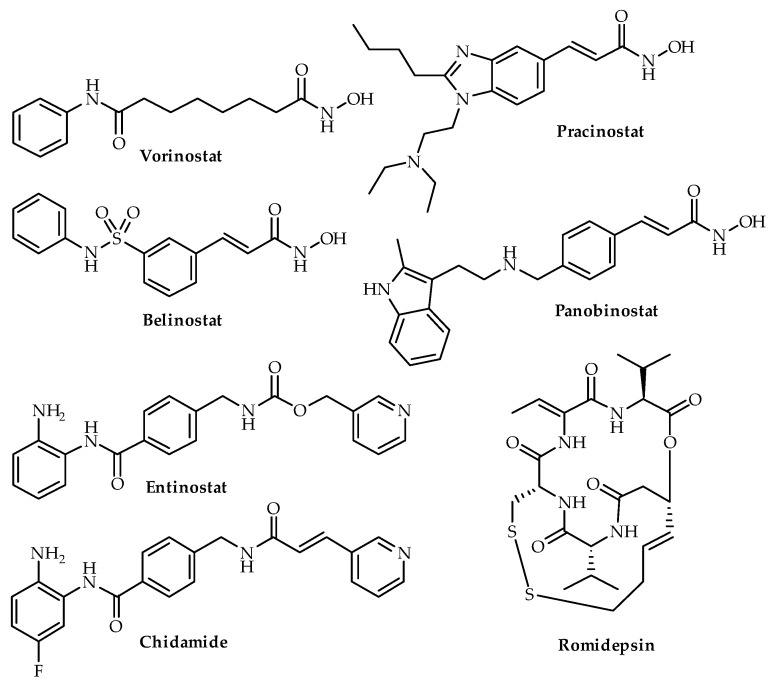
Chemical structures of four HDAC inhibitors (vorinostat, belinostat, romidepsin, pracinostat, entinostat, chidamide and panobinostat). Vorinostat, belinostat, romidepsin and panobinostat are FDA-approved.

**Table 1 biomedicines-08-00022-t001:** Localization and function of different classes of HDACs.

HDAC Class	HDAC Type	Localization	Function
I	1 /2	Nuclear	Interact with the promoters of vimentin, Slug and ZEB1 and upregulate EMT [22] Promotes DNA repair via NHEJ [23] Significant in stem cell functionality and clonogenic capacity in intestinal and embryonic stem cells [24,25] HDAC2, in complex with HDAC3 and KLF4 reduces VEGF expression, downregulating angiogenesis [26]
	3		Negative regulator of angiogenesis [27] TIP60 ubiquitination and cytoplasmic localization and protects cells from apoptosis after DNA damage [28]
	8		Involved in deacetylation of SMC3 leading to cohesion recycling during the cell cycle [29]
IIa	4	Nuclear/ cytoplasmic	Downregulation results in HIF-1α, MEKK2 and STAT1 acetylation [30,31,32] Induction of p21 inhibited by HDAC4 silencing allowing cells to progress from G1 to S phase [33]
	5		Interacts with the promoter of FGF2 and SLIT2, downregulating endothelial cell growth [34] Represses angiogenesis by downregulating FGF2 and Slit2, resulting in reduced sprout formation [35] Colocalizes with YY1, maintaining a terminally-differentiated state in cardiomyocyte cells (H92C) [36] Represses transcription of cyclin D3 [37] Required for nuclear accumulation of HIF-1α in hypoxic conditions [38] Promotes proliferation of tumor cells [39] Leads to reduced glucose uptake in primary muscle cells (with reduction in GLUT4 (*SLC2A4*) gene but increased GLUT1 expression) and insulin-stimulated glycogen synthesis [40]
	7		Upregulates PDGF-B, negatively regulating angiogenesis Represses MEF2 activity [41] Inhibits Stat3 activity by directly deacetylating Stat3, promoting lung tumorigenesis [42] Represses myeloid genes, preventing transdifferentiation of pre-B cells into macrophages [43] Augments the CSC phenotype of breast and ovarian cancer [44]
	9		Deacetylates ATDC, reducing ATDC-p53 interaction, and consequently inhibiting cell proliferation [45] Triggers gluconeogenesis by deacetylating FoxO1 and altering gene expression profiles of PGC-1α, cyclic AMP-responsive element-binding protein (CREB) and glucocorticoid receptor [46]
IIb	6	Nuclear/ cytoplasmic	Overexpression of HDAC6 in mammalian cells promotes chemotactic cell movement [47] Deacetylates HSP90, a chaperone protein, prolonging substrate protein action (e.g., AR) [48] Directly interacts with ubiquitin protein and dynein and transports misfolded proteins through microtubules to the perinuclear aggresomes. Polymers then degrade misfolded proteins by autophagy [49]
	10		Involved in DNA double-strand break repair in neuroblastoma cells [50] Promotes angiogenesis by deacetylating PTPN22 in endothelial cells [51] Functions as a polyamine deacetylase [52] Overexpression in HeLa cells triggers cellular DNA MMR activity specifically by deacetylating MSH2 [53]
IV	11		Inhibition results in a differential expression of genes involved in cytoskeleton remodeling, chromatin assembly and transcription, particularly WNT and PPAR-signaling pathways. Regulates survival genes in colorectal cancer patient (HMOX1), growth inhibition (GCNT3), apoptosis (AK2, TFAP2A) and differentiation of colorectal cancer stem cells (BMP4) [54]

**Table 2 biomedicines-08-00022-t002:** HDAC inhibitors undergoing clinical trials.

HDAC Inhibitor	Drug Combination	Tumor Type	Trial Phase Completed [Reference]
Entinostat	Exesmestane	Recurrent hormone receptor-positive breast cancer	Phase II trial [64]
	Permolizumab	Metastatic uveal melanoma	Phase II trial [65]
Chidamide	Exesmestane	Hormone receptor-positive, HER2-negative breast cancer	Phase III trial [62]
Vorinostat	Permolizumab	Advanced/metastatic Non-small cell lung cancer	Phase I trial [66]

**Table 3 biomedicines-08-00022-t003:** IC_50_ values of HDAC inhibition [57,71,72,73].

Compound	HDAC Inhibition IC_50_ (nM)										
	Class I				Class IIa				Class IIb		Class IV
	1	2	3	8	4	5	7	9	6	10	11
SAHA	38	144	6	38	>30000	>30000	>300000	>30000	10	21	28
Panobinostat	3	3	4	248	23	NA	18	6	3	ND	ND
Belinostat	41	125	30	216	115	NA	67	128	82	ND	ND
Pracinostat	49	96	43	140	56	47	137	70	1008	40	ND
Chidamide	95	160	67	733	>30000	>30000	>30000	>30000	>30000	78	432
Entinostat	262	306	499	2700	>30000	>30000	>30000	>30000	>30000	254	0.649
Mocetinostat	150	290	1660	>10000	>10000	>10000	>10000	ND	ND	ND	590
Romidepsin	36	47	ND	ND	510	ND	ND	ND	14000	ND	ND

ND = Not Determined.

**Table 4 biomedicines-08-00022-t004:** HDAC inhibitors examined in clinical trials to treat prostate cancer [10].

Name	Structural Class	HDACs Inhibited	Prostate Cancer Clinical Trial Status
SAHA	Hydroxamic acid	I, IIb and IV	II failed
Depsipeptide (Romidepsin)	Cyclic peptide	HDAC 1, 2 and 4	II failed
Pracinostat (SB939)	Hydroxamic acid	I, IIa and HDAC10	II failed
Panobinostat	Hydroxamic acid	HDAC I, 4, 6, 7 and 9	II failed

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
