# Peer review of "Understanding Failure and Improving Treatment Using HDAC Inhibitors for Prostate Cancer"

_biomedicines, 2020, doi:10.3390/biomedicines8020022_

Round 1
Reviewer 1 Report
Dear authors have adressed all my suggestions and consequently the review can be accepted for publication in Biomedicines
Reviewer 2 Report
According to my previous evaluation, the paper is suitable for publication.
This manuscript is a resubmission of an earlier submission. The following is a list of the peer review reports and author responses from that submission.
Round 1
Reviewer 1 Report
The review article of Rana et all focuses on HDAC inhibitors for prostate cancer therapy
In general, the review article is very-well written and of high interest. There are just some improvements necessary before the article can be accepted for publication in “biomedicines”.
Comments:
The numbering of the Figures and Tables is missing! Please insert an abbreviation list in the review article that would strongly facilitate reading. E.g. the abbreviation of CRPC is in the abstract, so if the reader is on page 4 he will not find the explanation for this abbreviation. Line 218: AR?, line 341: cyclosporine A (CsA); line 362: LBH? Please also draw the chemical structure of belinostat in figure 4 and depict in the figure which are approved and which are just under clinical trials. Please explain somewhere shortly the “Gleason score”. Although this review focusses on prostate cancer it would be important to include in the “HDAC inhibitors” section also other HDAC inhibitors (with their chemical structure) currently under clinical trials. Such as the benzamides: Tucidinostat, entinostat, mocetinostat etc. It is very likely that they are also tested against prostate cancer after the first approval in another (solid) cancer type. It seems that so far just the class of hydroxamic acids (and Romidepsin) has been tested for prostate cancer and not the class of benzamides. Any explanation why? Tucidinostat has been studied in a phase III trial against breast cancer (with positive results), so also a solid cancer type, this should be mentioned and discussed. Line 254: Why the name of this chapter is “side effects”. “Other cellular targets of HDAC inhibitors” would be better. The authors describe here that also other targets like VEGFR are inhibited. As the VEGFR inhibition itself is a very well-known target (with already approved drugs) this can even be a positive effect and not a negative “side effect”. For the conclusion it is possibly of interest to mention that >1000 zinc-containing enzymes are present in the human body. Consequently, it is not easy to selectively inhibit one class of zinc enzymes by e.g. a hydroxamic acid chelator moiety. Typing: Line 79: HDACGleason; line 185 “who were to be”; line 297: HDACRay; line362: LBHSpecifically; line 413: pAdministrationReviewer 2 Report
The review is well-organized and clearly presented. The topic is hot. There is really not much to add.